# Upskilling health and care workers with augmented and virtual reality: protocol for a realist review to develop an evidence-informed programme theory

Norina Gasteiger [1,2,3] Sabine N van der Veer [2,4] Paul Wilson [3]
Dawn Dowding [1]

For numbered affiliations see end of article.

**Correspondence to**
Norina Gasteiger;
norina.gasteiger@postgrad.manchester.ac.uk

## ABSTRACT

**Introduction** Augmented reality (AR) and virtual reality (VR) are increasingly used to upskill health and care providers, including in surgical, nursing and acute care settings. Many studies have used AR/VR to deliver training, providing mixed evidence on their effectiveness and limited evidence regarding contextual factors that influence effectiveness and implementation. This review will develop, test and refine an evidence-informed programme theory on what facilitates or constrains the implementation of AR or VR programmes in health and care settings and understand how, for whom and to what extent they 'work'.

**Methods and analysis** This realist review adheres to the Realist And Meta-narrative Evidence Syntheses: Evolving Standards (RAMESES) standards and will be conducted in three steps: theory elicitation, theory testing and theory refinement. First, a search will identify practitioner, academic and learning and technology adoption theories from databases (MEDLINE, Scopus, CINAHL, Embase, Education Resources Information Center, PsycINFO and Web of Science), practitioner journals, snowballing and grey literature. Information regarding contexts, mechanisms and outcomes will be extracted. A narrative synthesis will determine overlapping configurations and form an initial theory. Second, the theory will be tested using empirical evidence located from the above databases and identified from the first search. Quality will be assessed using the Mixed Methods Appraisal Tool (MMAT), and relevant information will be extracted into a coding sheet. Third, the extracted information will be compared with the initial programme theory, with differences helping to make refinements. Findings will be presented as a narrative summary, and the MMAT will determine our confidence in each configuration.

**Ethics and dissemination** Ethics approval is not required. This review will develop an evidence-informed programme theory. The results will inform and support AR/VR interventions from clinical educators, healthcare providers and software developers. Upskilling through AR/VR learning interventions may improve quality of care and promote evidence-based practice and continued learning. Findings will be disseminated through conference presentations and peer-reviewed journal articles.

## Strengths and limitations of this study

► Including quality assessments and identifying our confidence in each context, mechanism and outcome configuration will improve applicability of the programme theory.
► The repeated search will help to include recently published and up-to-date literature.
► This review will be conducted systematically, which enhances reproducibility.
► The literature review may be subject to selection bias because it will only include published, peer-reviewed studies in English.
► The mechanisms extracted will likely be untested and subjective author hypotheses.

## INTRODUCTION
### Upskilling in the health and care workforce
Upskilling through continuous learning and development is important in any business to improve skill sets, advance practice and close gaps in knowledge. Upskilling is the process of learning new skills or refining existing skill sets to enable employees to continue practising with ease.[1] For health support and care workers, upskilling ensures that their work is safe and aligns with best practice guidelines, as they often receive variable and inconsistent training, as non-registered staff.[2 3] Upskilling, in this sense, is therefore essential for providing consistent and high-quality care. Additionally, this promotes workforce flexibility and enables for the delegation of skills, when systems experience a shortage of staff.[4] Within the provision of health and care, upskilling is also crucial when adapting in times of change[5 6] or crisis[7] and to align with up-to-date best practice.

Health and care providers may range from registered clinicians such as surgeons, general practitioners and doctors, nurses and midwives, to allied health and non-registered

staff who provide care. Allied health staff may include paramedics, dieticians, podiatrists and radiographers,[8] while carers also include those working for care-based organisations such as in care homes or home care agencies. Regardless of the role, all staff that provide health and care services must act in accordance with policies/guidelines and optimally engage in up-to-date evidence-based best practice.

Evidence-based practice is widely recognised as the gold standard when providing effective and safe healthcare.[9] This requires professionals to update and upskill themselves on current evidence and to alter their practice to align with this, as well as with their patient's preferences.[10] Current evidence is usually retrieved from peer-reviewed journal articles; however, due to time constraints and workload demand, many health and care staff rely on organisational policies and protocols as formal sources of knowledge.[11] As the evidence base grows, old habits must be adapted and upskilling is required to align with the newest best practice.

Upskilling is also essential when adapting in times of change or crisis. For example, the emergence of medical and healthcare technologies requires staff to upskill, including improving their digital literacy skills.[5 6] Additionally, the novel COVID-19 pandemic caused significant changes to health and care systems. Changes included staff deployment to wards (eg, COVID-19 wards) outside of their normal experience and of retired and newly qualified staff, remote provision of healthcare using telehealth (phones, video, patient portals), distancing/minimal contact care, stringent use of personal protective equipment and strengthened interprofessional collaboration.[12–15] These challenges required prompt upskilling, especially in using technologies and in infection prevention and control behaviours to minimise the spread of COVID-19.

### Upskilling training programmes

Upskilling training programmes traditionally consist of e-learning, textbooks, workshops, seminars, shadowing/observation and reading peer-reviewed journal articles. Hatfield et al[16] systematically reviewed 12 studies that delivered behaviour change training interventions to healthcare professionals. All used educational elements (eg, presentations and workshops) and most were delivered in person. Morris et al[17] reviewed training interventions aimed at carers. Both reviews concluded that interventions that use both educational and practical elements (eg, practising skills or discussion) are most effective.[16 17] This indicates that education-only interventions may not be effective in upskilling health and care staff.

Time, organisational structure, difficult to access resources and a reliance on experiential knowledge also constrain providers from upskilling.[3 11] Health and care staff have widely reported a preference for learning through 'doing' (such as interacting with or observing colleagues), rather than from journal articles or textbooks.[18–20] Additionally, although support and care

workers provide clinical, care-based and clerical patient care, their value is often not reflected in their allocated training budgets and available programmes.[3] As a result, many feel insufficiently prepared.[3] However, clinical, health support and care staff indicate a willingness to upskill, receive further training and to participate in interventions that will improve their practice.[11 21] Further, some managers and nurses in England-based nursing homes have expressed enthusiasm towards implementing innovative digital health technologies that may improve residents' quality of care.[22]

### Digital technologies for upskilling

Effective interventions that are short, accessible, interactive, memorable and low cost are needed to overcome training barriers. For workplaces with staff shortages, training also needs to be flexible and provided on a drop-by basis.[3] Brief interventions delivered via digital technology may be appropriate, as they can be made available online and accessed 24/7. They can also be more engaging and memorable by including interactive activities (eg, games, quizzes, simulations and immediate performance feedback). However, there is limited literature on implementation strategies for digital interventions that upskill health and care workers. Theories of change can be applied to knowledge of existing barriers and facilitators to using digital health programmes for healthcare workers. Lewin describes behaviour as 'a dynamic balance of forces working in opposing directions'.[23] Lewin theorises that driving forces (ie, facilitators) and restraining forces (ie, barriers) counter one another, but can result in change if one over-rides the other. This means that barriers and facilitators directly impact the implementation success and effectiveness of digital training programmes for health and care staff.

Literature on digital health technologies has highlighted various driving and restraining forces that impact both implementation and the effectiveness of programmes. Keyworth et al[24] conducted a review of 69 studies to determine what maximises the effectiveness and implementation of technology-based interventions that support healthcare professional practice. They concluded that successful technologies employ behaviour change theories and specific instruction on how to perform behaviours. They also provide professionals with knowledge and person-specific information to assist with practice (eg, patient management). Driving forces for implementation include integration into clinical workload, alignment with organisational strategies and senior peer endorsement. Restraining forces include organisational challenges, as well as the design, content and technical issues of the interventions.

Literature also highlights key strategies for implementation, focusing on provider adoption and acceptance. Recommendations for facilitating change include linking new practice with old practice to build familiarity,[25 26] identifying people who are willing to facilitate and promote the new practice[26 27] and to clearly communicate to staff

as to how the new practice will benefit them and their patients.[26 28 29] Spagnoletti *et al*[28] provide specific examples, highlighting that short sessions, role-modelling content (eg, video clips of the behaviour) and modules that refresh understanding of familiar curriculum were important in their implementation of an online training programme for interns.

## Simulation technologies for upskilling

The implementation of simulation technologies may be a novel and engaging approach to upskilling health and care workers. The term health and care workers captures the breadth of professionals working in health and social care, including medical staff, general practitioners, nurses, carers and community workers. Simulation in this context refers to the replication of real-life interactions or scenarios, whereby learners receive immediate feedback/debriefing.[30] Various levels of simulation exist, depending on 'fidelity' (reality). According to Seropian *et al*,[31] these can be categorised as high, medium and low fidelity and use tools such as human-like body parts, haptic feedback, computer programs (eg, serious games) or virtual reality (VR) headsets to facilitate experimental learning. Low-fidelity simulation may include a simple body part, such as a doll-like arm to practise intravenous insertion skills.[32] In contrast, high-fidelity simulation tools include real-life responses driven by computers.[32] These are more expensive and may include the METI Human Patient Simulator, which looks and acts like a human (eg, blinks, has a pulse and speaks) and accurately mirrors responses to clinical procedures, such as intubation and catheterisation. However, it is important to note that simulators mimic, rather than replicate reality.[32]

Simulation technology has been found to be as effective as traditional teaching methods for educating health and care staff and students.[33–35] However, when compared with traditional methods, students report better retention of knowledge[36] and higher satisfaction and motivation when using simulation technologies such as games.[34] Experimental learning by simulation also allows for learners to repeatedly practise skills and make and learn from their mistakes without harming a patient, distressing them or facing other negative consequences.[32 37] Computer-driven simulation technologies such as games, augmentation and VR also enable independent learning, often without the need for an instructor to immediately provide feedback or debrief learners. Debriefing can then occur at a later date, such as to determine trainee performance and learning progress.

In VR, users wear a headset to become immersed in a digital environment. Headsets range from the low-cost Samsung Gear VR or Google Cardboard to high-end gaming equipment such as Oculus Touch. The extent of immersion also differs, ranging from non-immersion (eg, using computer-based VR), semi-immersion and fully immersive simulations (eg, those with haptic feedback). The perception of being immersed within a non-physical world is created through various stimuli, including images and sound,[38] which enable users to learn from experience. In interactive medical VR, users can engage in virtual worlds, including with patients and colleagues, and react to specific scenarios.[30] In contrast, within augmented reality (AR), real-world environments are complemented with interactive computer-generated imagery and information.

Unlike traditional simulators, the main benefit of VR is transporting the learner into an immersive environment. VR and AR interventions are also cost-effective as they can be used autonomously, independently and repeatedly, compared with traditional simulation technologies. In fact, they have been deemed as the learning tool of the 21st century[39] and their popularity is expected to continually increase. Current projections for the AR/VR head-mounted display market include a worth of US$25 billion by 2022, with an annual growth rate of 39.5%.[40] This highlights that now is the ideal time to research implementation of AR/VR due to an inevitable growth in use and further reduction in costs.

These technologies have transformed clinical training and have been used to support healthcare workers in decision-making and teaching emergency response, resuscitation, robotic surgery and alcohol screening skills.[41–45] However, their effectiveness is contested within the literature, with some research stating that VR is not as effective as other training tools, including for phlebotomy training.[46] Other literature highlights that VR is useful for 'presence', but does not improve learning outcomes.[47 48] It is hypothesised that VR increases cognitive load and therefore compromises cognitive resources from the learning experience.[47] Conversely, some research has found VR to be more effective than other educational techniques,[49 50] with systematic reviews concluding that VR training is effective in improving technical skills for arthroscopic surgery[51] and knowledge and skill performance when learning clinical psychomotor skills.[52] Evidently, research is needed to explore to what extent and for whom VR interventions are effective.

Despite their contested effectiveness, VR and AR technologies have now been commercialised and implemented to upskill and support health providers. FundamentalVR,[53] for example, provides flight simulator-like training for surgeons with the use of haptic elements for tactile feedback. In the SentiAR[54] tool, holographic visualisations are provided for each patients anatomy and float alongside or above the patient during procedures (eg, treating cardiac arrhythmias). Other tools include the AR xVision[55] three-dimensional anatomical images that enable clinical providers to see a patient's skin and tissue (akin to X-ray vision) and the AR SureWash[56] mobile app, which provides personalised feedback for hand hygiene technique.[57] VR technologies were also implemented during the COVID-19 pandemic when face-to-face teaching was not possible.[58] For example, St Bartholomew's Hospital used VR to train their nurses and doctors on 50 clinical procedures.[59] Their OMS VR

system provided performance feedback, tracked improvement and facilitated group learning.

## Gap in research and aim

Despite the emergence and potential efficacy of simulation technologies, the effectiveness of these technologies as an educational intervention remains debated. This includes how good they are at enabling upskilling compared with other strategies, and how they can be implemented into a practice setting, to enable upskilling. Additionally, as evident in the mixed findings on the effectiveness of AR and VR interventions in upskilling staff, programme interventions, including digital ones, do not work for everyone equally.[60] A gap in research remains on the factors that influence when an AR or VR intervention works, to what extent, for whom and in which context. Moreover, research is needed on the causal mechanisms that influence the outcomes of AR/VR interventions and their implementation. This is essential in ensuring that future digital interventions are designed and appropriately targeted at health and care workers for both maximum efficiency and sustained effects. The aim of this review is to develop, test and refine an evidence-informed programme theory on what facilitates or constrains the implementation of AR or VR programmes in health and care settings and understand how, for whom and to what extent they 'work'.

## METHODS AND ANALYSIS
### Realist review

This research will take a realist approach because it can produce useful answers to complex questions often left unexplored by experimental research.[60] These questions include: how, when, for whom and to what extent does an intervention 'work'? To answer these questions, realist approaches consider the complex interactions between the environment, individuals and the intervention.

Realist evaluation is an emerging theory-driven methodology that seeks to understand CMO configurations, that is, the context (C), mechanisms (M) and outcomes (O) of interventions. Context refers to the backdrop of conditions that may impact outcomes, such as organisational structure, functional fidelity, environmental settings, culture and norms. These trigger or modify mechanisms (causal forces) that influence outcomes.[61] Examples of mechanisms include the resources offered by interventions or changes in reasoning or behaviour.

Realist reviews seek to understand context, mechanisms and outcomes by identifying candidate theories and then systematically reviewing literature for underlying social entities, processes or social structures that result in the intended outcome[62]; rather than assuming that the intervention itself produces an outcome. This process is useful for complex interventions, in which outcomes may not necessarily be linear, and instead depend on the context and both intended and unintentional mechanisms.[62] It also allows exploring how an intervention is meant to

work compared with how it actually works in practice.[63] Additionally, 'demi-regularities' are identified to acknowledge that outcomes will vary across contexts, but some CMO patterns will remain.[61] This focuses reviewers on the transferable aspects of a programme theory.[62] By definition, candidate theories are individual and specific theories, while a programme theory provides an overarching explanation of how a specific intervention is expected to work, including how contexts and mechanisms lead to negative and positive outcomes.[64]

CMO configurations are then developed as a programme theory, which is tested and refined in real-life settings and with key stakeholders.[60] As with AR/VR technologies, the main benefit of realist evaluation is the ability to bridge theory and practical application in the contexts and with the populations that the intervention targets.[60]

A realist review will therefore help to answer the following questions:

- ► What facilitates or constrains the implementation of AR/VR programmes in health and care settings?
- ► What are the mechanisms by which VR/AR interventions result in their intended outcomes?
- ► What contexts determine whether the different mechanisms produce their intended outcomes?
- ► In what circumstances and for whom are VR/AR interventions effective in upskilling health and care providers?

The core research team is a multidisciplinary group of researchers from the backgrounds of nursing, primary healthcare, health informatics and implementation. Across this group, expertise relevant to the topic includes that on digital health innovation and evaluation, behaviour change, implementation science and conducting realist reviews. The Realist And Meta-narrative Evidence Syntheses: Evolving Standards (RAMESES) training documents[62] will be referred to, and the review will be reported in accordance with the RAMESES publication standards for realist synthesis[65] (online supplemental table S1).

## Procedures

Realist reviews tend to follow a three-step process: theory elicitation, theory testing and theory refinement. This process will be followed to describe our procedures. Unlike systematic reviews, which aim to uncover all research relevant to the topic, realist reviews find a comprehensive balance of empirical research and theory.[66] Searches will therefore be iterative and additional rounds of searching may alter the following procedures. Figure 1 highlights the processes that will be conducted in each stage.

### Theory elicitation
#### Search strategy

A search will be conducted to identify initial candidate theories. These will not be limited by publication date and are characterised as academic, practitioner and learning and technology adoption theories.

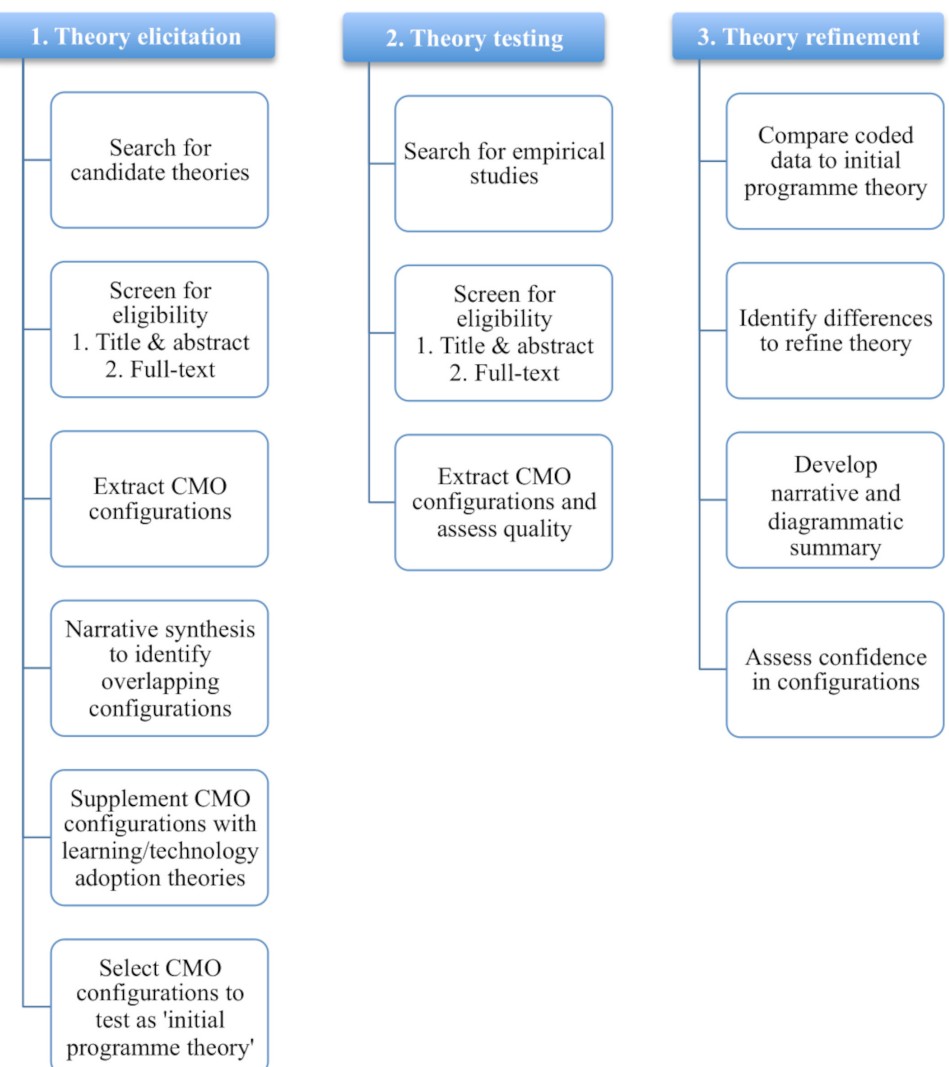

**Figure 1** Summary of the three steps and processes that will be conducted. CMO stands for context, mechanism and outcome.

We will identify academic and practitioner theories using free text and Medical Subject Headings terms when searching MEDLINE, Scopus, CINAHL, Embase, Education Resources Information Center, PsycINFO and Web of Science. Snowballing will also help to identify relevant work. Online supplemental table S2 provides the search strategy. An initial search of the databases in January 2021 located 811 items, of which 200 were deemed potentially eligible, after reviewing their titles and abstracts.

We will focus on the discussion section of items to identify why AR or VR interventions did or did not achieve their intended outcomes. These often include the author's theories.[67] Existing systematic reviews will first be reviewed.

Relevant practitioner theories may be presented by professional bodies, or within grey literature, including editorials, letters, news articles and commentaries.[68] We will therefore supplement the above search with the additional journals presented in table 1. Literature from other relevant journals such as the Journal of Medical Internet Research and the Journal of the American Medical

Informatics Association will be identified through the database searches, as they are indexed in MEDLINE.

We will also identify theories related to adult learning from the academic articles. These are expected to include theories related to adult learning for health and care professionals, including Schön's[69] theory on the reflective practitioner and Slotnick's[70] theory on how physicians learn. Two theories identified by Mukhalalati and Taylor[71] as key to professional learning are directly applicable to AR/VR. These include:

► Experiential learning, whereby knowledge construction and learning are facilitated through interaction with the environment. Kolb[72] proposes a framework for experiential learning that includes concrete experience, reflective observation, abstract conceptualisation and active experimentation.

► In constructivism, learning occurs through interaction between previous skills/knowledge, those gained through social interaction and social activities, within the learner's environment, physical and social world.[73] Simulation has been identified as a tool that supports

**Table 1** Summary of relevant journals related to continued learning in health and care

| Professional body | Journal/s |
| --- | --- |
| Association of American Medical Colleges | Academic Medicine; MedEdPORTAL |
| Association for Medical Education in Europe | Medical Teacher |
| Foundation for Advancement of International Medical Education and Research | International Journal of Medical Education |
| Alliance for Continuing Education in the Health Professions; Association for Hospital Medical Education; Society for Academic Continuing Medical Education | Journal of Continuing Education in the Health Professions |
| German Association for Medical Education | GMS Journal for Medical Education |
| The Australian & New Zealand Association for Health Professional Educators | Focus on Health Professional Education |
| Association for the Study of Medical Education | Medical Education |
| **Journals not associated with a professional body** | |
| Journal of Nursing Education and Practice Nurse Education Today International Journal of Nursing Studies | |

constructive learning,[74 75] as constructivists generally believe that people learn best by 'doing', as this is how they construct their knowledge.[76]

Theories may also relate to technology acceptance and adoption. Frameworks include:

► The Diffusion of Innovations theory,[77] which explains how and at what rate innovations (eg, technologies) spread, as determined by different categories of adopters. This can be applied to organisations and individuals.
► The Technology Acceptance Model[78] explains that an individual's perceived usefulness and ease of use of a technology influence intention to use and actual use.
► The Unified Theory of Acceptance and Use of Technology[79] determines that four constructs: performance expectancy, effort expectancy, social influence and facilitating conditions (eg, age, gender, experience and voluntariness of use) influence an individual's technology use and acceptance.
► The Non-adoption, Abandonment, Scale-up, Spread, Sustainability framework[80] evaluates reasons for non-adoption, abandonment and challenges to implementation through six domains (condition, technology, value proposition, adopter system and institutional and societal contexts).

► The Consolidated Framework for Implementation Research[81] considers five domains related to the intervention, outer and inner settings, the individuals involved and implementation process.
► The Normalisation Process Theory[82] focuses on people's actions, rather than their intentions/beliefs. It considers coherence, cognitive participation, collective action and reflexive monitoring as crucial to the implementation process.

### Record management
Similar to the methods in Randell *et al*'s[63] study, records will be saved to an EndNote library, as well as charted on Excel. A timeline sheet on Excel will record search activities, including the databases searched, the date of each search and the number of records found.

### Screening
Two researchers (NG and DD) will screen the literature for eligibility, starting by determining the relevance from the title and abstract, and then reading the full text. As in other realist reviews, the first researcher will screen all items and generate a shortlist of possible eligible items, while the second researcher independently screens a random subset of items (20%) at each screening stage.[83] A raw agreement rate will be calculated to determine inter-rater reliability, while any disagreements will be resolved through discussion, so that consensus is met. The inclusion criteria for the academic and practitioner theories will be:

► Using simulation technologies (any type of immersion will be accepted).
► Health and care workers and individuals postgraduation/registration as learners.
► Any health, care or university-based setting (as these often have simulation labs).
► Includes detail on implementation and/or on what contexts, how and for whom they 'worked'.
► Published in English.

The exclusion criteria include simulation technologies that do not use augmentation or VR (eg, low-fidelity web-based e-learning interventions or manikin-only simulators), undergraduate students and published in languages other than English. Work also including undergraduate learners or other simulation technologies will only be included if the data for postgraduate/registered learners and AR/VR can be separated. Undergraduate students will be excluded as they differ from learners postregistration. Namely, they are learning content for the first time, rather than upskilling their clinical or practical knowledge/experience. For the purpose of this review, VR is defined as a computer-generated simulated environment, while AR refers to the projection of computer-generated imagery onto real-world environments.[84 85]

A Preferred Reporting Items for Systematic Reviews and Meta-Analyses (PRISMA) flow chart[86] will document the review selection and decision process.

> **Box 1  Content to be extracted from included sources and recorded in the coding sheet**
>
> ► Author; date.
> ► Title.
> ► Type of publication (journal paper, conference paper or book chapter).
> ► Research design, theoretical orientation (if applicable) and methods.
> ► Augmented reality/virtual reality (AR/VR) technology description.
> ► Study objective (focus).
> ► Setting; country.
> ► Sample (type, size, age, gender).
> ► Context.
> ► Mechanism.
> ► Outcome (intended, unintended and/or subjective).
> ► Implementation (strategy, adoption and/or uptake).
> ► Learning or technology adoption theories mentioned (if applicable).

### Analysis and synthesis

We will extract relevant information (presented in box 1) including that pertaining to context, mechanism and outcomes from each article from the academic and practitioner theories. Adult learning and technology adoption theories will be briefly summarised. For consistency, outcomes should broadly be related to the Kirkpatrick[87] components of evaluation: reaction (ie, satisfaction), learning (ie, knowledge), behaviour or results (skills). Unintended and other subjective or observed outcomes (eg, increased confidence or perceived interactivity) will be included too. A second reviewer will code and extract data from a random selection of 10–20% of the articles to ensure consistency in interpretation.

All information will be recorded in an Excel sheet for analysis. If possible, complete CMO configurations will be recorded, however; it is unlikely that all articles will contain complete statements—fragments will therefore be recorded too.[63]

On completion, we will conduct a narrative synthesis to determine any overlapping CMO configurations. These will then be compared with identified (learning and adoption) theories to further explore the underlying causal mechanisms so as to understand how VR/AR interventions can or should upskill health and care professionals.[88] The resulting CMO configurations will answer: (A) What facilitates or constrains the implementation of AR/VR programmes in health and care settings? (B) How, for whom and to what extent did they produce the intended outcomes (reaction/satisfaction, short-term and long-term learning/knowledge and behaviour/results)?

The research team will then select a number of CMO configurations to test, focusing on those that are most feasible and likely to apply to future AR or VR interventions.

### Theory testing
### Search strategy

We will search databases to identify empirical and published studies that will enable the CMO configurations

to be tested. First, we will identify the empirical literature from step 1. We will then search the same databases as in step 1, using the same keywords, but limit the timeframe of the search to only include recently published literature that we will have missed since conducting the first search. Snowballing will help to identify additional literature. This will consist of checking the reference lists of the included literature.

### Screening

The articles will be screened by determining their relevance to the programme theory (eg, AR/VR tools used by health and care workers). A benefit of realist reviews is the focus on the intervention mechanism, enabling the inclusion of literature whereby the intervention has been applied to different settings, people or even similar interventions in the same setting.[68 89] All study designs will be included. A PRISMA diagram will visualise the study selection process.[86]

### Analysis and synthesis

Relevant information (presented in box 1) will be extracted into an Excel sheet. We will also assess the quality of each paper using the Mixed Methods Appraisal Tool (MMAT), as this is appropriate for qualitative, quantitative and mixed methods research.[90] The MMAT was developed in 2007,[91] and revised in 2011.[92] Unlike earlier versions, the newest 2018 MMAT is not intended to be quantified and instead offers a guide for discussing quality. We will therefore highlight methodological flaws to inform recommendations for future research. Low-quality research will not be excluded, as realist methodologists acknowledge that useful information on contextual factors may be present.[93] In alignment with the guidelines for conducting realist reviews, the quality of each study will focus on the evidential fragment (relevant section) that each theory is drawn from.[93] For example, when only quantitative data are used from a mixed methods study to test the theory, the quality of the qualitative component will not be assessed. Cohen's kappa will be calculated to determine inter-rater reliability between the two authors conducting the quality assessments.

### Theory refinement

Coded data will be compared with the initial programme theory, and differences will be identified to refine and revise the programme theory. On completion of the final theory, a narrative and diagrammatic summary will be presented.[64 94] We will use the MMAT to assess the extent to which we are confident in each finding. Ultimately, each CMO configuration will be rated as high, moderate, low or very low in confidence. This rating will highlight areas for research and also support decision-makers when deciding whether to implement or develop similar technologies to upskill health and care workers.

### Strengths and limitations

Inherent limitations of realist reviews must be acknowledged. Realist reviews have been critiqued to be laborious

and time intensive,[95] so the included literature is not always up to date when it is published. We will overcome this through a second database search, which will specifically identify recently published work. Programme theories are also only as good as the literature they include, but they do sometimes not acknowledge or assess quality.[83] We are therefore conducting quality assessments of the literature and using this to identify our confidence in each CMO configuration. A fundamental limitation we cannot overcome but must acknowledge is that mechanisms are often untested and subjective author hypotheses,[96] which may limit the accuracy of the programme theory.

### Patient and public involvement
Members of the public were not involved in the development of this protocol.

### ETHICS, DISSEMINATION AND CONCLUSION
Ethics approval is not required to conduct this realist review. This protocol describes how we will conduct a realist review that constructs, tests and refines an evidence-informed programme theory on what facilitates or constrains the implementation of AR/VR programmes in health and care settings and how, for whom and to what extent they 'work'. The results may inform and support AR/VR interventions from clinical educators, healthcare providers and software developers. Upskilling through AR/VR learning interventions may ultimately improve quality of care and promote evidence-based practice and continued learning. Findings will be disseminated through conference presentations and peer-reviewed journal publications. In our future work we will continue to refine our programme theory by involving stakeholders. This will include interviews as well as experimental work.

**Author affiliations**
¹Division of Nursing, Midwifery and Social Work, Faculty of Biology, Medicine and Health, The University of Manchester, Manchester, UK
²Division of Informatics, Imaging and Data Sciences, Centre for Health Informatics, Faculty of Biology, Medicine and Health, The University of Manchester, Manchester, UK
³Division of Population Health, Health Services Research and Primary Care, Faculty of Biology, Medicine and Health, The University of Manchester, Manchester, UK
⁴Manchester Academic Health Science Centre, The University of Manchester, Manchester, UK

**Contributors** NG conceived and designed the study with support from DD, SNvdV and PW. NG wrote the first draft of the manuscript. All authors revised and approved the final manuscript.

**Funding** This work is funded by the National Institute for Health Research Applied Research Collaboration Greater Manchester.

**Disclaimer** The views expressed in this publication are those of the authors and not necessarily those of the National Institute for Health Research or the Department of Health and Social Care.

**Competing interests** None declared.

**Patient consent for publication** Not required.

**Provenance and peer review** Not commissioned; externally peer reviewed.

**ORCID iDs**
Norina Gasteiger http://orcid.org/0000-0001-7801-7417
Sabine N van der Veer http://orcid.org/0000-0003-0929-436X
Paul Wilson http://orcid.org/0000-0002-2657-5780
Dawn Dowding http://orcid.org/0000-0001-5672-8605

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
