## [Reviewer comments · BMJ Open]

ARTICLE DETAILS

TITLE (PROVISIONAL)	Upskilling health and care workers with augmented and virtual reality: Protocol for a realist review to develop an evidence-informed programme theory
AUTHORS	Gasteiger, Norina; van der Veer, Sabine N.; Wilson, Paul; Dowding, Dawn

VERSION 1 – REVIEW

REVIEWER	Osmanliu, Esli CHU Sainte-Justine, Pediatric Emergency Medicine
REVIEW RETURNED	26-Mar-2021

GENERAL COMMENTS	This study aims to understand how contextual factors influence the outcomes of augmented or virtual reality-based upskilling among health and care workers. The authors will perform a realist review to address this objective. They will present findings through a narrative summary. I congratulate the authors for highlighting the relevance of the research question and for submitting a thoughtfully developed protocol. The following comments may help further strengthen the protocol. In particular, I recommend a deeper discussion of the study's limitations. Please note that the line (l.) numbers below refer to the second numbered column in the manuscript (the one closest to the text). Abstract, l.35: The discussion of study limitations is superficial. Can the authors think of other limitations than the ones listed, particularly with respect to inherent limitations of a realist review? p.5, l. 7: Why is the phrase "health support and care workers" better suited than more typical formulations such as "health care worker" or provider? Please highlight the advantages of the former - is it to more inclusively capture the breadth of professionals working in health and social services? p.7, l. 4: Consider discussing the role of functional fidelity, beyond a focus solely on structural fidelity p7, l. 24: It seems premature to suggest that feedback is not necessary with such educational technologies; feedback may still play an important role even if delivered asynchronously or via pre-established triggers based on trainee performance.
---

	p.7, l.39: Why can't it be both an immersive and educational environment? p.8, l.26: the claim that there is absence of research on AR/VR effectiveness contradicts the two prior paragraphs. Did the authors mean that the effectiveness of these technologies as educational interventions remains debated? p.9: Please discuss the main limitations and criticism of realist reviews, and how you plan to overcome those. p.9: Did you plan to register your protocol to an online registry, in addition to submitting to this journal? PROSPERO may not accept realist review protocols but the Open Science Framework (OSF) is an alternative. p.10, l.25: Additional sources to consider include JMIR, JAMIA p.12, l.4: Will you provide some measure of inter-rater reliability? Will the two researches review the same body of literature or share the work? p.12, l.14: Why exclude undergrad students? Couldn't theories originate from studies in this population and still be informative for your target population? p.12, l.17: Please include a formal/technical definition of AR and VR to standardize results. p.12, l.42: Would it also be pertinent to gather information on the degree of skill retention and contextual factors that affect those? p.13, l.5: Can the authors describe precisely how they will approach the snowballing technique, in order to promote replicability. p.13, l.40: A critical appraisal of this protocol from health and care providers as well as other stakeholders (e.g. policy makers; administrators; program implementers) may provide useful insight to the research team. This could occur through interviews and/or focus groups and determine in part whether the questions posed are reflective of current needs, comprehensive, and actionable. Similarly, a critical review and interpretations of the study findings with a similar group can be helpful. These steps are left to the discretion of the authors. p.20: Which categories will you use for publication type?
--	---

REVIEWER	Meum, Torbjorg University of Agder
REVIEW RETURNED	22-Apr-2021

GENERAL COMMENTS	This paper presents a protocol for a realist review on the use of AR or VR programs in health and care settings. The motivation and purpose of the study is clearly described in the introduction which provides and overview of the use of digital training programs in health care. The study is based on previous research on the use of VR/AR in trainings programs and the author argue well for the need to explore the causal mechanisms that influence the outcomes of AR/VR interventions. Furthermore, the method
--

	section is well described and all the steps in the planned review process are outlined in accordance with the guidelines for conducting realist reviews. Overall, I find the paper interesting, and I believe that the planned realist review has the potential to contribute to increased knowledge about the use of AR or VR in training programs. I have just a few comments that authors can take into consideration in the final stage of submission. Realist evaluation is a theory-driven methodology and theory development, testing and refinement are well described in the paper. However, I think the term programme theory can be defined more clearly to explain the difference between programme theories and candidate theories. The search strategy is described as an iterative process and two search strategies are described (search for candidate theories and search for empirical studies). However, the same databases and keywords will be used in both searches. I therefore recommend a clarification of how the second search strategy is a follow-up and extension of the first search strategy. As mentioned, these are just minor recommendations and all in all I think the review process is well described.
--	---

VERSION 1 – AUTHOR RESPONSE

Reviewer: 1

Dr. Esli Osmanliu, CHU Sainte-Justine

This study aims to understand how contextual factors influence the outcomes of augmented or virtual reality-based upskilling among health and care workers. The authors will perform a realist review to address this objective. They will present findings through a narrative summary.

I congratulate the authors for highlighting the relevance of the research question and for submitting a thoughtfully developed protocol.

The following comments may help further strengthen the protocol. In particular, I recommend a deeper discussion of the study's limitations. Please note that the line (l.) numbers below refer to the second numbered column in the manuscript (the one closest to the text).

Thank you!

Abstract, l.35: The discussion of study limitations is superficial. Can the authors think of other limitations than the ones listed, particularly with respect to inherent limitations of a realist review?

We have amended our limitations (page 2). They now include:

- The literature review may be subject to selection bias, because it will only include published, peer-reviewed studies in English.
- The mechanisms extracted will likely be untested and subjective author hypotheses.

We have also added a short paragraph on page 12 (lines 32-42) that acknowledges the inherent limitations of realist reviews: Inherent limitations of realist reviews must be acknowledged. Realist reviews have been critiqued to be laborious and time-intensive[95], so the included literature is not always up-to-date when it is published. We will overcome this through a second database search,

which will specifically identify recently published work. Programme theories are also only as good as the literature they include, but they do sometimes not acknowledge or assess quality[83]. We are therefore conducting quality assessments of the literature and using this to identify our confidence in each CMO configuration. A fundamental limitation we cannot overcome but must acknowledge is that mechanisms are often untested and subjective author hypotheses[96], which may limit the accuracy of the programme theory.

p.5, l. 7: Why is the phrase "health support and care workers" better suited than more typical formulations such as "health care worker" or provider? Please highlight the advantages of the former - is it to more inclusively capture the breadth of professionals working in health and social services?

We have clarified this on page 5, lines 7-9. We state: The term health and care workers captures the breadth of professionals working in health and social care, including medical staff, general practitioners, nurses, carers and community workers.

p.7, l. 4: Consider discussing the role of functional fidelity, beyond a focus solely on structural fidelity

We have now acknowledged functional fidelity as a contextual factor that may trigger mechanisms (page 7, line 7): Context refers to the backdrop of conditions that may impact outcomes, such as organisational structure, functional fidelity, environmental settings, culture and norms.

p7, l. 24: It seems premature to suggest that feedback is not necessary with such educational technologies; feedback may still play an important role even if delivered asynchronously or via pre-established triggers based on trainee performance.

We have re-framed this statement (page 5, lines 27-29), to highlight that feedback/de-briefing does not need to occur immediately, but can still be important at a later date to monitor progress and performance. This says: Computer-driven simulation technologies such as games, augmentation and VR also enable independent learning, often without the need for an instructor to immediately provide feedback or de-brief learners. De-briefing can then occur at a later date, such as to determine trainee performance and learning progress.

p.7, l.39: Why can't it be both an immersive and educational environment?

We have removed 'rather than educational one' on page 5, lines 41-42. This sentence now reads: Unlike traditional simulators, the main benefit of VR is transporting the learner into an immersive environment.

p.8, l.26: the claim that there is absence of research on AR/VR effectiveness contradicts the two prior paragraphs. Did the authors mean that the effectiveness of these technologies as educational interventions remains debated?

Yes- that is what we meant! We have changed our wording to reflect this (page 6, lines 28-29). Despite the emergence and potential efficacy of simulation technologies, the effectiveness of these technologies as an educational intervention remains debated.

p.9: Please discuss the main limitations and criticism of realist reviews, and how you plan to overcome those.

We have added a paragraph on page 12 (lines 32-42) that acknowledges the inherent limitations of realist reviews, and highlights the strengths and limitations of our work (including how we will overcome them).

Strengths and limitations

Inherent limitations of realist reviews must be acknowledged. Realist reviews have been critiqued to be laborious and time-intensive[95], so the included literature is not always up-to-date when it is published. We will overcome this through a second database search, which will specifically identify recently published work. Programme theories are also only as good as the literature they include, but they do sometimes not acknowledge or assess quality[83]. We are therefore conducting quality assessments of the literature and using this to identify our confidence in each CMO configuration. A fundamental limitation we cannot overcome but must acknowledge is that mechanisms are often untested and subjective author hypotheses[96], which may limit the accuracy of the programme theory.

p.9: Did you plan to register your protocol to an online registry, in addition to submitting to this journal? PROSPERO may not accept realist review protocols but the Open Science Framework (OSF) is an alternative.

Realist reviews are often not formally registered. As the main purpose of registering a protocol is so the work is not duplicated and to provide a deeper explanation of the methods, we believe that publishing our protocol open access in BMJ Open will help us to achieve the same objectives. We therefore do not plan to register the protocol.

p.10, l.25: Additional sources to consider include JMIR, JAMIA

We have added a statement on page 8, lines 26-28 that addresses these journals: Literature from other relevant journals such as JMIR and JAMIA will be identified through the database searches, as they are indexed in Medline.

p.12, l.4: Will you provide some measure of inter-rater reliability? Will the two researchers review the same body of literature or share the work?

We have now clarified the screening process and interrater reliability calculation on page 10 (lines 14-18): As in other realist reviews, the first researcher will screen all items and generate a short-list of possible eligible items, while the second independently screens a random sub-set of items (20%) at each screening stage[83]. A raw agreement rate will be calculated to determine interrater reliability, while any disagreements will be resolved through discussion, so that consensus is met.

We have also clarified this on page 12 (lines 19-20): Cohen's kappa will be calculated, to determine interrater reliability between the two authors conducting the quality assessments.

p.12, l.14: Why exclude undergrad students? Couldn't theories originate from studies in this population and still be informative for your target population?

We have clarified this on page 10, lines 30-32. We now state: "Undergraduate students will be excluded as they differ from learners post-registration. Namely, they are learning content for the first time, rather than upskilling their clinical or practical knowledge/experience."

p.12, l.17: Please include a formal/technical definition of AR and VR to standardize results.

On page 10 (lines 32-34), we have now added definitions for AR/VR: For the purpose of this review, VR is defined as a computer-generated simulated environment, while AR refers to the projection of computer-generated imagery onto real-world environments[84,85].

p.12, l.42: Would it also be pertinent to gather information on the degree of skill retention and contextual factors that affect those?

On page 11 (line 15) we have clarified that the outcomes focussed on include short and long-term outcomes: (b) How, for whom and to what extent did they produce the intended outcomes (reaction/satisfaction, short and long-term learning/knowledge and behaviour/results)?

p.13, l.5: Can the authors describe precisely how they will approach the snowballing technique, in order to promote replicability.

On page 11, lines 28-29, we now explain that: This will consist of checking the reference lists of the included literature.

p.13, l.40: A critical appraisal of this protocol from health and care providers as well as other stakeholders (e.g. policy makers; administrators; program implementers) may provide useful insight to the research team. This could occur through interviews and/or focus groups and determine in part whether the questions posed are reflective of current needs, comprehensive, and actionable. Similarly, a critical review and interpretations of the study findings with a similar group can be helpful. These steps are left to the discretion of the authors.

Thank you for your suggestion. In our future work we will continue to refine our programme theory, by involving stakeholders. This will include interviews, as well as experimental work. The researchers involved in developing this protocol and in conducting the review also have relevant practical and academic backgrounds, including clinical nursing, implementation of digital health interventions, public health and health informatics.

We have clarified this on page 13 (lines 9-11): In our future work we will continue to refine our programme theory, by involving stakeholders. This will include interviews, as well as experimental work.

p.20: Which categories will you use for publication type?

We have clarified this in Box 1 on page 11. We state: Type of publication (journal paper, conference paper or book chapter).

Reviewer: 2

Dr. Torbjorg Meum, University of Agder

This paper presents a protocol for a realist review on the use of AR or VR programs in health and care settings. The motivation and purpose of the study is clearly described in the introduction which provides an overview of the use of digital training programs in health care. The study is based on previous research on the use of VR/AR in training programs and the authors argue well for the need to explore the causal mechanisms that influence the outcomes of AR/VR interventions. Furthermore, the method section is well described and all the steps in the planned review process are outlined in accordance with the guidelines for conducting realist reviews. Overall, I find the paper interesting, and I believe that the planned realist review has the potential to contribute to increased knowledge about the use of AR or VR in training programs. I have just a few comments that authors can take into consideration in the final stage of submission.

Thank you!

Realist evaluation is a theory-driven methodology and theory development, testing and refinement are

well described in the paper. However, I think the term programme theory can be defined more clearly to explain the difference between programme theories and candidate theories.

We have added a clearer definition to distinguish them (page 7, lines 19-22), we state: By definition, candidate theories are individual and specific theories, while a programme theory provides an overarching explanation of how a specific intervention is expected to 'work,' including how contexts and mechanisms lead to negative and positive outcomes[64].

The search strategy is described as an iterative process and two search strategies are described (search for candidate theories and search for empirical studies). However, the same databases and keywords will be used in both searches. I therefore recommend a clarification of how the second search strategy is a follow-up and extension of the first search strategy. As mentioned, these are just minor recommendations and all in all I think the review process is well described.

We have added an explanation of how the second search is an extension of the first, on page 11 (lines 25-28). We state: First we will identify the empirical literature from step one. We will then search the same databases as in step one, using the same keywords, but limit the timeframe of the search to only include recently published literature that we will have missed since conducting the first search.

We have also added this to our abstract (page 2, line 20): Second, the theory will be tested using empirical evidence located from the above databases and identified from the first search.

VERSION 2 – REVIEW

REVIEWER	Osmanliu, Esli CHU Sainte-Justine, Pediatric Emergency Medicine
REVIEW RETURNED	28-May-2021
GENERAL COMMENTS	I congratulate the authors for effectively incorporating suggestions from the initial review process.
REVIEWER	Meum, Torbjorg University of Agder
REVIEW RETURNED	10-Jun-2021
GENERAL COMMENTS	The authors have made a great effort to respond to my comments and I believe that the paper is ready for publication. The paper is well written and provides a thorough description of the planned realist review. I believe that the study protocol addresses a highly relevant topic and I look forward to reading the results of this study